# The Use of Infographics to Inform Infection Prevention and Control Nursing Practice: A Descriptive Qualitative Study

**DOI:** 10.3390/healthcare13222961

**Published:** 2025-11-18

**Authors:** Susana Filipe, Maria Manuel Borges, Amélia Castilho, Celeste Bastos

**Affiliations:** 1Health Sciences Research Unit: Nursing (UICISA: E), Nursing School of Coimbra (ESEnfC), 3046-851 Coimbra, Portugal; afilomena@esenfc.pt; 2Local Health Unit of Baixo Mondego, 3094-001 Figueira da Foz, Portugal; 3CEIS20, University of Coimbra, 3004-530 Coimbra, Portugal; mmb@fl.uc.pt; 4CINTESIS@RISE, Nursing School of Porto (ESEP), 4200-072 Porto, Portugal; cbastos@esenf.pt

**Keywords:** continuing education, infection control, infographics, information dissemination, nursing, qualitative research, epidemiologic surveillance, visual communication

## Abstract

**Background**: Healthcare-associated infections surveillance is a cornerstone of infection prevention and control, essential for guiding effective program implementation. Timely and clear dissemination of surveillance data is critical to inform decision-making and improve clinical practice. However, traditional lengthy reports are often overlooked due to time constraints among healthcare professionals. To address this, a secondary hospital introduced infographics as a concise and visually engaging method to communicate healthcare-associated infections surveillance data, aiming to enhance feedback and promote more impactful communication. This study explores infection prevention and control link nurses’ perspectives on using infographics to disseminate healthcare-associated infections surveillance data. **Methodology**: A descriptive qualitative design was employed, using semi-structured individual interviews conducted with thirteen purposively selected participants. Narrative data were analyzed using Bardin’s content analysis methodology. **Results**: Three key themes emerged as follows: Data awareness; infographic use; and team engagement. Participants emphasized that infographics simplified complex data, improved accessibility, and fostered team engagement. When integrated into educational and clinical workflows, infographics were perceived as effective tools for promoting reflection and supporting a culture of continuous quality improvement. **Conclusions**: Our findings suggest that infographics offer a promising approach to enhance communication of surveillance data. Their use may support informed decision-making and contribute to improved infection prevention and control practices.

## 1. Introduction

Surveillance of Healthcare-Associated Infections (HAIs)—infections acquired during care that were neither present nor incubating at admission—is a cornerstone of infection prevention and control (IPC) programs, aimed at improving clinical practice and reducing avoidable infections [1]. It involves the systematic collection, analysis and interpretation of healthcare data to inform IPC strategies [2]. Timely feedback of surveillance findings to all stakeholders is essential [1], yet its effectiveness depends not only on the data itself but also on how it is communicated. As Borges [3] notes, “the value of data lies in its use”, underscoring the importance of accessible and engaging formats for data dissemination.

Infographics have emerged as a preferred format for presenting complex information, offering advantages over text-only summaries by reducing cognitive load and enhancing comprehension [4]. This is particularly relevant for IPC teams, who routinely produce HAIs surveillance reports for hospital stakeholders. As Scott et al. [5] suggest, presenting information in a more engaging, visible and memorable format—such as infographics—can improve both the reach and impact of these reports, making complex data more accessible and actionable for diverse audiences.

Infographics combine text and visuals to convey information quickly and clearly [6,7]. While often seen as a modern communication tool, their use has historical roots. Florence Nightingale’s “polar area diagram”, for example, illustrated the causes of mortality among British soldiers during the Crimean War and played a key role in public health reform [8]. Later, Peter Sullivan’s infographics for *The Sunday Times* helped popularize visual storytelling in journalism, improving information accessibility [9].

Today, infographics are widely used across sectors—including social media, advertising, policy and academic publishing—as a compelling strategy for conveying complex messages [10]. In healthcare, they have gained recognition as tools capable of enhancing the delivery of information to both patients and professionals [5]. Specifically, they support patient health literacy [11,12], improve medical communication [13,14], and bolster competencies in evidence-based practice [15]. Within nursing education, infographics have demonstrated utility as instructional resources in domains such as pharmacology, in care philosophy and basic life support training [16,17,18]. To our knowledge, their use in professional nursing practice and information-conveyance to practicing nurses remains underexplored.

Despite the growing adoption of infographics, there is a paucity of research addressing nurses’ perspectives on the deployment of infographics to disseminate surveillance data on HAIs. The present study therefore aims to explore the perspectives of IPC link nurses regarding the use of infographics to communicate HAI surveillance findings. Further details on the development of the infographics are described in Section 2.3 to provide full transparency on the materials used in this study.

## 2. Materials and Methods

### 2.1. Research Design

A descriptive qualitative study was conducted to address the aim of this study. Semi-structured interviews were selected as the data collection method, given the possibility for deeper insights into participants’ perspectives while maintaining alignment with the study objectives. This approach offers a balance between flexibility and structure, particularly valuable for capturing nuanced, context-specific views [19].

The research questions were developed following a review of the relevant literature on infographic usage in healthcare communication and the dissemination of surveillance data among nursing professionals. This review revealed limited evidence regarding the views of practicing nurses on the use of infographics for healthcare-associated-infection (HAI) feedback. Based on these identified gaps, and in alignment with the study’s aim to explore the utility of infographics among infection prevention and control (IPC) link nurses, we formulated three research questions: (1) Are IPC link nurses aware of HAIs surveillance data? (2) What are IPC link nurses’ perceptions of using infographics to disseminate HAIs surveillance data? (3) Do IPC link nurses believe that infographics have impacts on data feedback and awareness of infection prevention and control?

### 2.2. Setting and Participants

This study was conducted in a 170-bed secondary hospital and involved nurses serving as IPC link professionals. These participants were purposively selected due to their dual function: acting as intermediaries between clinical teams and the IPC team, and simultaneously representing the nursing staff within their respective units. Their position enables them to play a pivotal role in the translation of IPC policies into practice, while also providing insight into the educational needs, challenges, and facilitation issues experienced by frontline nursing staff in the implementation of IPC strategies.

Participants were eligible if they held an IPC link nurse role, were actively employed in the hospital setting, and expressed willingness to participate. Nurses who did not satisfy these inclusion criteria were considered ineligible.

Participants were approached in person and invited to take part in the study. Upon acceptance, interviews were scheduled at their convenience. All participants received both written and verbal information about the study, and written informed consent was obtained before data collection. Confidentiality was maintained by anonymizing all transcripts and removing any identifying information such as names, workplaces, or personal details.

### 2.3. Background

Aligned with one of the World Health Organization’s (WHO) core components for an IPC program [1], HAIs surveillance reports are biannually issued to ward directors and head nurses, and annually to the hospital board and all healthcare professionals. These reports are supplemented by meetings with IPC link professionals and head nurses to reinforce key messages.

Despite these efforts, it remained unclear whether frontline nurses were consistently receiving and engaging with the information, raising concerns about the limitations of traditional dissemination methods. To address this, an infographic (Figure 1) was introduced as a complementary communication tool, aiming to present HAIs data in a format that was more engaging, relatable, and easier to interpret for nursing staff and other healthcare professionals.

The infographic design followed six key principles, as proposed by Murray et al. [20]: (1) tailoring content to the target audience—healthcare professionals; (2) using a compelling title, in this case incorporating wordplay; (3) providing a clear narrative structure; (4) emphasizing key messages; (5) balancing images, charts and text; and (6) limiting the color pallet and font variety.

Following its initial development, and consistent with recommendations by Arcia et al. [21], the infographic was informally reviewed by a random sample of five healthcare professionals. Their feedback informed refinements to the design, ensuring clarity and relevance of the content. From the researchers’ perspective, this iterative process was essential to validate the infographic’s effectiveness as a communication tool.

### 2.4. Data Collection

A semi-structured interview guide was developed to align with the study’s research questions and explore IPC link nurses’ perceptions of infographics for healthcare-associated infection (HAI) surveillance data dissemination. The development process involved a review of the relevant literature on infographic usage and data feedback practices, followed by the drafting of open-ended questions and prompts. Although a formal pilot test of the interview guide was not conducted due to time/resource constraints, it was reviewed by a qualitative research expert and refined accordingly. The final guide covered key domains: awareness of HAI surveillance data; perceptions of infographic-based dissemination; experiences of current feedback methods; and perceived impacts of the infographics. When participants provided brief responses, we followed up with pre-listed prompts—used flexibly—such as, “Could you tell me more about that?”, and “How did you experience that situation?” The complete guide (including main questions and probes) is provided as Appendix A.

Data collection occurred between September 2023 and January 2024. Eleven interviews were performed face-to-face in a private meeting room, and two were held via videoconference (Zoom^®^) due to scheduling constraints. Interviews lasted between 40 and 75 min, with a total of 13 interviews conducted. After the 10th interview, no new substantive codes or themes were identified in the final three interviews, confirming that data saturation had been achieved. All interviews followed the same guide and were conducted by the first author. Each session was audio-recorded and transcribed verbatim. After transcription, the interview transcripts were shared with the participants (i.e., member checking) to ensure accuracy and allow for clarifications or corrections. Codes and themes generated from the data were subsequently confirmed through triangulation with other researchers, enhancing the trustworthiness of the analysis. The researcher had no direct or hierarchical professional relationship with the participants.

### 2.5. Data Analysis

Data were analyzed following Bardin’s [22] content analysis methodology, which comprises three sequential phases: (1) pre-analysis; (2) exploration of the material; and (3) treatment of results and interpretation.

All interviews were transcribed in full. In the pre-analysis phase, a floating reading of the transcripts was conducted to promote familiarization with the data and to identify preliminary impressions and directions for analysis [22].

Transcripts were then segmented into meaning units that captured significant content. These units were analyzed and, where appropriate, grouped into thematic categories. The coding process was guided by the researchers’ interpretative lens, informed by critical reflection and expertise in infection prevention and control. A semantic criterion was applied, whereby content was grouped according to shared meanings or ideas, supporting coherent thematic interpretation [22]. Codes were generated inductively from the data.

In the final phase, coded units and categories were reviewed, refined, and validated in relation to the study objectives. The interpretative analysis focused on identifying patterns, frequencies, and relationships among categories, considering the broader context. Codes with similar meanings were consolidated into broader categories, and related categories were clustered to form overarching themes [22]. The final step involved defining and naming these themes, which were organized into a matrix table to provide a structured overview.

Data were analyzed using AIQDA^®^ software (https://aiqda.com/), enabling segmentation of verbatim statements and facilitating the coding process.

To guide the reporting of this study, we followed Consolidated Criteria for Reporting Qualitative Research (COREQ) [23] (checklist available as Appendix A).

Data convergence across participants confirmed the robustness of emerging categories and the absence of new relevant information, reinforcing the evidence of saturation. Rigor was maintained through the study design and methodological procedures. All interviews were conducted, transcribed, and reviewed by the same researcher to ensure trustworthiness. Credibility was enhanced through validation of emerging codes by the research team, and confirmability was supported by the inclusion of participant quotations.

### 2.6. Ethical Considerations

All ethical considerations were respected. Authorization was obtained from both the Hospital Board and the Ethics Committee of the same institution, which issued a favorable opinion (code 14.OBS.2023, 7 August 2023). Free and informed consent was obtained from all participants, who were informed of their right to withdraw from the study at any time without consequence. Participation was entirely voluntary and conducted in accordance with the ethical principles outlined in the Declaration of Helsinki [24].

## 3. Results

A total of thirteen participants were included in the study. All were female, aged between 30 and 52 years (M = 39.5 years, SD = 6.4 years), and had between 1 and 14 years of experience in the IPC link nurse role (M = 6.0 years, SD = 3.9 years), representing a broad spectrum of professional expertise.

Through narrative analysis, coded units and categories were reviewed, refined, and validated in relation to the study objectives, resulting in three main themes: (1) Data awareness—reflecting participants’ views on current communication methods and their impact on practice; (2) infographic use—focusing on the perceived benefits of infographics in terms of accessibility, readability and visual appeal; and (3) team engagement—exploring the need to involve broader nursing teams and promote collective responsibility for HAIs surveillance. Figure 2 presents these themes alongside their respective categories and subcategories, providing a structured overview of the findings.

The selected verbatim statements were labeled with the number assigned to each participant, enclosed in brackets.

### 3.1. Data Awareness

#### 3.1.1. Importance of Data Awareness

All participants expressed awareness of HAIs surveillance data and recognized its importance in fostering professional reflection, highlighting the impact of clinical practices. This awareness was seen as a trigger for self-assessment and improvement. Participants noted that when data reveals less favorable outcomes, it prompted both individual and collective reflection on professional responsibilities within the institution. Moreover, openly sharing these results was perceived as a means to promote deeper accountability and encourage meaningful dialog among teams.

*They are important (referring to data) because they make us reflect on our practices (…) if we see that the results are not so good, it might lead us to reflect that we, as professionals of an institution, may need to do better (…). Bringing it out to the open, for awareness, I think it leads us to that, to reflection*.(P1)

*I think it has everything to do with awareness, because the more we know, the more we become aware*.(P12)

Participants also highlighted the role of surveillance data in strengthening the connection between clinical practice and outcomes. The data was seen not only as a diagnostic tool, but also as a catalyst for change, guiding nurses toward informed decision-making and improvement. There was also a shared understanding that data provides direction; it shows where the institution has been, where it stands, and where it aims to go. This sense of progression supports strategic planning and encourages a culture of learning and evolution.

*With this data we have tools to try and make a difference, to improve, for the coming years. Because we know where we have been, where we are, and where we want to go. And with this kind of documents, we can understand our evolution*.(P2)

*I think that only data makes us change the way we work, to improve (…) What can be changed? That’s the importance (…) it kind of brings us back to reason*.(P6)

Moreover, participants acknowledge that data serves as evidence, either confirming that practices are being correctly implemented or revealing gaps that need attention. This connection between practice and evidence reinforces the value of surveillance as a foundation for evidence-based nursing.

*They are important (referring to data) because they are evidence that either we are doing things correctly in terms of practice care, or, really, something is failing in our practice (…) the connection between practice and evidence ends up being demonstrated through the results*.(P11)

#### 3.1.2. Dissemination Methodologies

Participants emphasized that disseminating surveillance data on HAIs through infographics represented a marked improvement over previous methods. They consistently pointed out that accessing long reports on computers posed challenges, primarily due to time constraints and dependence on individual initiative, which could be compromised by workload and competing priorities.

Participants noted that although surveillance data were technically available, it often did not reach the teams in a meaningful and actionable way.

*Between having a report with 20 or 30 pages, and having an infographic, I think it’s much easier to see something that is simple and explicit than, for example, going to the compute or flipping through a document... you know, you lose some time reading. One thing is having a report (…) another thing is having the information in a strategic place where you can go and see the information*.(P1)

*The information gets to the ward, but then, (…) the way it is shared with the team, the one’s that really matter… well, it is said: ‘There is information on the intranet, check it out because these data are important for you to see’. Given the workload people have, and sometimes even their interest, not everyone does it (check the intranet), obviously*.(P13)

In contrast, infographics were perceived to enhance visibility and accessibility, offering a clear and organized overview of HAI data. This format allowed nurses to quickly grasp essential information and share it easily with peers, promoting team-wide awareness and engagement.

*(…) less time-consuming for nurses, because they don’t have to go through a document with many pages (…) here (referring to the infographic) we have all the information, organized, easy to interpret, and then it’s easy to pass it on to others, to peers*.(P2)

*(...) that’s when the information really reached us, because until then, maybe the data was available, but we wouldn’t go looking for it, and there was no visibility. And this way, it came to us*.(P3)

*(…) the way the results are shown to the teams, through images. That is important. In the past, it was just the written report that was sent, and there you go, sometimes it didn’t get to the team. It was difficult*.(P13)

Overall, the interviews revealed that while traditional reporting methods posed barriers to access, the introduction of infographics facilitated a more inclusive and practical approach to data dissemination.

#### 3.1.3. Data Awareness Impact

Participants’ reflections illustrate how the infographic served not only as a tool for disseminating surveillance data but also as a catalyst for professional reflection and action. Participants described a heightened awareness of infection trends and a renewed motivation to uptake their knowledge, reassess clinical practices, and initiate ward-level improvement projects.

*(…) for knowing that there was an increase in central venous catheter associated infections, it even led me to do more research on the topic (…) it kind of pushed me to look into it, to update myself*.(P1)

*The fact that I saw the incidence of urinary tract infection in our hospital made me rethink this practice and what needs to be changed to improve this infection rate. Because being in practice, I see the complications this brings to the patients*.(P7)

*(…) the data on respiratory infections that you put there, ‘Are we neglecting oral hygiene?’ caught my attention (…) and being able to connect everything, it’s definitely an important improvement in my ward*.(P8)

The visual and accessible format of the data appeared to bridge the gap between information and practice, empowering nurses to engage critically with their environment and take ownership of change.

*It allows us to create improvement projects, within our own wards, by seeing what we’re really not doing so well, and so, focus on the problem and implement actions to correct it*.(P10)

### 3.2. Infographic Use

#### 3.2.1. Accessibility of Information

Participants described the use of infographics as a notably effective strategy for enhancing the accessibility and engagement of HAIs surveillance data within clinical settings. The visual format was perceived as intuitive and timesaving, enabling nurses to swiftly comprehend key information without the need to consult lengthy reports.

*(…) through the posters (the infographics) people have easy access, they don’t need to go looking anywhere because it’s right there, visible, and if they’re interested … it’s in front of us, we don’t have to do anything, just be interested*.(P3)

*It’s even more perceptible than actually reading the report, isn’t it? It takes less effort, you look at the diagram and all the information is there in a systematic way. I think that, without a doubt, this might be the main reason why people now give more value, or pay more attention, in this case, to the report data (…) it’s a very innovative way, it’s easily perceptible and visible*.(P11)

*(…) these latest forms of information, I think they were easier, it was an easier way to reach colleagues. Because it’s something that isn’t hidden and that we don’t have to go looking for. And so people are more likely, even if just to look and see what it is, and they end up reading it and sometimes even getting interested in finding more information. So, I think there was an evolution, really a way to reach team members*.(P13)

This ease of access was seen as particularly valuable in fast-paced environments, where time and attention are limited.

*We live in a time where we no longer have time. We no longer have time for deep reflections, we want to avoid complexity, having more work. Infographics, in a simplified way, really allow us to implement solutions (…) this really speaks to us, it truly appeals to our common sense, honestly, and we immediately get the idea (…) it’s about wanting to improve*.(P10)

Several participants noted that the visibility of the infographic—its placement within the ward—played a crucial role in whether it captured attention and encouraged interaction. When strategically displayed, participants believe that the infographic is more likely to spark curiosity and prompt further exploration of the data.

*(…) I didn’t like that it was placed behind the door. I don’t think it’s an appropriate spot*.(P8)

*It needs to be clearly visible (…) it has to be displayed so that colleagues can notice and see it. The location is very important. It’s not behind a door that we just stick it*.(P13)

Additionally, the integration of a QR code was highlighted as a meaningful enhancement, offering a seamless way to access the more detailed institutional report, when needed.

*Then, since it has the QR code, we can download the Institution’s entire report*.(P10)

*(…) the information has to be delivered in an objective way. It has to be really concise because, considering all the hustle we deal with and the concerns related to the ward, it needs to be something very concise. It has to be there and ready. And if we have any doubts, we can access through the QR code, and that helps a lot*.(P13)

By improving both the format and the visibility of information, infographics help overcome the common barrier of nurses not actively seeking out surveillance data. Additionally, the possibility of complementing these visuals with more detailed information is a valuable asset, allowing for deeper understanding when needed.

#### 3.2.2. Readability and Interpretability

Two key concepts that emerged from the interviews were readability and interpretability, both of which play a critical role in making complex surveillance data accessible and actionable for nursing professionals. Participants emphasized that the infographic format was more inviting and less demanding, allowing for quick and effective engagement even during busy clinical routines. The clarity and organization of the information were seen as essential for facilitating understanding and enabling peer-to-peer communication. 

*It’s a method that, in my opinion, is more inviting to read*.(P1)

*We’re able to read it in a summarized and easy way (…) here we have all the information organized, easy to interpret, and then it’s easy to pass it on to others, to our peers*.(P2)

The participants appreciated being able to absorb relevant content in a short amount of time, highlighting how the concise and structured presentation helped the message come across clearly.

*It’s something we can do in the meantime, in a little bit of time we may have during worktime, it’s quick to read (the infographic), easy to interpret, it has the most relevant and most important information. I think that’s what makes it easier and makes the message get across*.(P7)

#### 3.2.3. Visual Design

Participants expressed appreciation for several visual elements of the infographic, including the color palette, the information layout, and the overall graphic style. These design choices were seen as contributing to a more inviting and engaging experience. The use of color was particularly noted for its ability to highlight key information and support visual filtering, while the inclusion of graphs and diagrams added clarity and appeal.

*It’s simple and clearly laid out with inviting colors*.(P1)

*The colours, they’re ok, it’s not all one colour. There’s something that stands out, so it helps us filter out some more relevant information. And presenting graphs also helps*.(P3)

The simplicity and coherence of the design were valued for making the content approachable and easy to absorb.

*The colour palette is pleasant (…) It’s visually attractive and so, you can immediately get a summary and everyone sees it*.(P10)

*(…) even the images in the diagrams end up being appealing. They could just be lines, but no, everything has its purpose and its reason. And I think that is, without a doubt, an added value*.(P11)

While most participants responded positively to the visual design, a few emphasized the importance of sobriety and simplicity, cautioning against excessive use of color or visual clutter.

*(…) I think that if we mix too many colours, it also gets a bit confusing. I don’t know… it becomes noisy*.(P1)

*(…) the simpler, the better*.(P5)

Another aspect highlighted was the integration of reminders and reflective prompts within the infographic. These elements were seen as enhancing comprehension and reducing cognitive load by encouraging critical thinking and connecting data to practice. Participants noted that such features made the infographic not only informative but also thought-provoking and action-oriented.


*The reflections, at the end of the data, the reflections (…) I think sometimes make us think a bit about things (…) It’s essentially that question—What can we do, isn’t it? It’s not just about looking at the data. Are we neglecting something?*
(P8)

*(…) and the reminders on the side, that are placed there* (on the infographic) *and that make us cross-reference the information to what is the most important.*(P10)

Overall, participants valued the infographic’s visual design for its ability to enhance understanding, reduce cognitive effort, and stimulate reflection, all of which contribute meaningfully to educational strategies in nursing practice.

### 3.3. Team Engagement

#### 3.3.1. Motivation

Although participants themselves were aware of HAIs surveillance data, they noted that most frontline nurses were not. This gap in awareness was attributed not to a lack of information availability, but rather to limited engagement and interest among staff. Participants emphasized the need for broader staff engagement beyond the IPC link nurses, pointing to a general lack of motivation or curiosity from some colleagues.

*I think that sometimes there isn’t much interest in knowing the data. It’s not that it isn’t provided, because it is, but sometimes I don’t really see much interest*.(P3)

*(…) the knowledge or lack of knowledge of each professional depends on what we set out to do (…) you can send the reports, you can give access to all kinds of documents, but there has to be interest from the professionals*.(P6)

These reflections suggest that while communication strategies are essential for disseminating HAIs data, their effectiveness ultimately depends on the motivation of healthcare professionals. Without genuine interest in engaging with the information, even well-designed tools may fall short in reaching their full educational potential.

#### 3.3.2. Complementary Strategies for Dissemination

All participants recognized the added value of complementing the dissemination of HAIs surveillance data with additional strategies, aimed at increasing reach and engagement. These included the use of digital tools, team involvement in dissemination decisions, and in-service training initiatives. One participant suggested adapting the infographic into a mobile-friendly format, such as an app, to make the information more accessible and tailored to healthcare professionals’ routines.

*If the infographic is transformed into some kind of information delivered through the mobile phone or an app, specific to the healthcare professional, who can access it in a schematic way like the paper infographic, but in the form of an app*.(P11)

To improve visibility and relevance, another participant proposed involving staff in decisions about where infographics are displayed. This participatory approach was seen as a way to foster ownership and ensure the materials are effectively positioned.

*(…) to know the general opinion, like using a mini-survey to know what people became aware of, or whether the location chosen was appropriate, if they were aware of it, or not (…) sometimes we might think it’s in the most suitable place, but maybe it’s not*.(P1)

Regarding in-service training, participants emphasized the importance of informal and integrated learning opportunities. These could include short sessions during shift handovers or targeted discussions led by IPC link nurses, with focus on ward-specific data.

*(…) at the end of the shift handover, dedicate a little time to the infographic and explain what it is, what is the objective, what are the results (…) even the IPC link nurses should take a more active role in sharing this information*.(P3)

*Elevate the infographic and explain (…) I would like to have access to the data from the ward, my ward in particular*.(P8)

Participants also acknowledge their own responsibility in promoting team engagement. One participant reflected on the proactive role IPC link nurses should take in facilitating dissemination.

*As soon as the results came out, we could do a training session, a short training session. The IPC link nurses should do it*.(P4)


*Maybe that data could be shared with the rest of the team, maybe the failure also lies a bit with the IPC link nurses not doing it, or at least trying to get those alerts across, right?*
(P12)

However, participants also recognized the challenges faced by IPC link nurses in fulfilling these responsibilities, particularly the lack of dedicated time to work with surveillance data and coordinate with the IPC team.

*The IPC link nurses need dedicated time to be able to work this data within the wards themselves, and act more effectively and in close connection with the IPC team. And I think that from there, things would be disseminated differently*.(P10)

These reflections highlight the importance of multimodal dissemination strategies that combine digital innovation, participatory decision-making, and structured educational efforts. Such approaches were seen as essential to enhancing awareness and fostering a culture of shared responsibility within healthcare teams.

## 4. Discussion

This study explored IPC link nurses’ perspectives on the use of infographics to disseminate HAIs surveillance data. In addressing the predefined research questions, the findings also revealed additional perspectives that deepen understanding of the topic. To facilitate interpretation, our discussion is organized according to the key themes that emerged from the data.

### 4.1. Data Awareness

While IPC link nurses were familiar with the epidemiological surveillance data, they observed that awareness among frontline nursing staff remained limited, highlighting an ongoing challenge in effective data dissemination. This gap did not result from a lack of data availability, as surveillance reports were accessible, but rather from low levels of staff engagement and interest. These findings underscore the persistent challenge in identifying and implementing optimal strategies for communicating surveillance data within clinical environments.

According to the WHO’s frameworks for infection prevention and control, effective HAIs surveillance requires not only the widespread distribution of reports but also ensuring that shared data prompts meaningful action from all stakeholders, including frontline professionals [1,25]. Passive information sharing, where reports are distributed without follow-up or facilitation, is insufficient on its own. Instead, data must be targeted to individuals and groups who both understand the implications and are empowered to act upon it.

Awareness of epidemiological surveillance is essential for evidence-based practice [25], enabling nurses to interpret and apply relevant data in real-world contexts. However, realizing this potential depends on ensuring that the information reaches frontline nurses effectively, ensuring that those directly involved in patient care are informed and engaged. As Mitchell and Russo [2] emphasize, surveillance is only effective when communication is actionable, that is, when it prompts responses and prevention in practice [2,26].

Participants emphasized the importance of team engagement and identified IPC link nurses as key facilitators in promoting responsibility for data dissemination—an established core function within infection prevention and control programs [1]. They further advocated for complementary strategies, such as digital tools and in-service training, to enhance uptake and promote a shared commitment to quality improvement. Despite the growing use of digital technology in nursing education, its integration into clinical settings remains limited, particularly for reinforcing surveillance data and supporting practical decision-making [27]. In contrast, in-service training continues to be a reliable method for reinforcing infection prevention and control guidelines and connecting data to everyday practice, as highlighted in WHO’s recent infection prevention and control training curriculum [28].

Participants consistently expressed a preference for disseminating HAIs surveillance data through infographics, rather than traditional reporting formats. This preference was attributed to the enhanced accessibility, visibility, and clarity that infographics offer, particularly in fast-paced clinical environments. However, promoting team-wide engagement was identified as a priority for future investment.

Reflecting on previous dissemination methods, IPC link nurses highlighted several advantages of infographics, including ease of access, visual appeal, and readability. These features facilitated communication with the broader nursing team. This aligns with findings from Scott et al. [5], who emphasize that in saturated information environments, infographics succeed by capturing attention and conveying information clearly.

Participants also noted that infographics reduce cognitive workload, a benefit supported by Patel et al. [14], who argue that healthcare professionals often lack the time to engage with lengthy reports. However, as Ibrahim [29] cautions, infographics—also referred to as visual abstracts—can offer a concise and accessible summary of key information, but should not replace full reports, as they do not capture all details. Participants echoed this concern, emphasizing the importance of access to complete surveillance reports, which they obtained via the QR code embedded in the infographic. They regarded this access as essential for fostering deeper understanding, supporting informed decision-making, and encouraging reflection on clinical practice. According to the literature, this dual strategy—using infographics to spark interest and linking them to full reports—reflects best practice in visual communication. As Ibrahim [29] emphasizes, the primary function of infographics is to attract attention and spark interest, which is particularly valuable in healthcare communication. However, this must be complemented by access to detailed information to avoid misinterpretation and support accurate clinical reflection.

### 4.2. Infographic Use

Infographics were perceived by participants as valuable educational tools that promote reflection, foster connections between clinical actions and outcomes, and support evidence-based decision-making. Their effectiveness was attributed to thoughtful visual design, readability, and strategic placement, which collectively reduce cognitive effort and enhance the accessibility of complex information.

Participants widely endorsed the use of infographics, appreciating their simplicity and clarity. Presenting health information using compelling visuals—such as charts, icons, and graphic elements—can mitigate information overload by providing concise and accessible overviews, as demonstrated in previous research [9]. As John Berger, cited in Siricharoen & Siricharoen [9], noted in *Ways of Seeing* (Penguin, 1972), “seeing comes before words”, reinforcing the idea that visual elements can enhance comprehension and retention. The participants’ perspectives suggest that integrating visual data tools into communication strategies can enhance nursing education and quality improvement initiatives within clinical settings by foresting critical thinking and proactive engagement with clinical challenges. Moreover, incorporating infographics and other visual tools into educational and clinical workflows has been shown to improve data literacy and make surveillance information more interpretable and actionable [25].

Building on the role of infographics in enhancing data accessibility and promoting behavioral change, it is important to consider the broader impact of visual perception on decision-making. Vision, as one of the most dominant and influential sensory modalities, plays a critical role in how individuals process and engage with information [30]. The findings of Thorndike et al. [31] further illustrate the impact of visual cues on human behavior. Their study implemented a color-coded food labeling system and reorganized cafeteria layouts to enhance the visibility of healthier options. This intervention demonstrated the effectiveness of choice architecture in promoting healthier consumption within a hospital setting. Similarly, a nudge intervention combining the strategic placement of alcohol-based hand rub with a visual reminder at patient’s bedsides significantly increased compliance with hand antiseptics [32].

Participants indicated that dissemination of surveillance data would benefit from complementary strategies extending beyond infographics alone. This reflects the importance of multimodal dissemination strategies that combine digital innovation, participatory decision-making, and structured educational efforts. Such approaches are essential for enhancing awareness and fostering a culture of shared responsibility within healthcare teams. This aligns with the WHO’s definition of a multimodal strategy, which comprises several elements—such as system change, training, reminders, and feedback—implemented in an integrated manner to improve outcomes and drive behavioral changes [1]. Given its versatility, implementing a multimodal approach in disseminating HAIs surveillance data to support sustained behavioral change presents a promising direction for future research.

### 4.3. Team Engagement

Participants reported that infographics positively influenced both data feedback and awareness of infection prevention and control practices and outcomes. They emphasized that increased data awareness promotes reflection, fosters critical thinking, and strengthens the ability to link nursing practice with patient outcomes—thereby deepening engagement with complex information and supporting more evidence-informed care.

Participants highlighted that accessibility to data—enabled by visual tools—facilitated the connection between practice and outcomes. These reflections align with the concept of data literacy, which encompasses the ability to interpret, evaluate, and apply data in clinical decision-making [33]. In the context of nursing education, fostering data literacy is essential for evidence-based practice. When integrated into educational and clinical workflows, infographics can make surveillance data more interpretable, actionable, and engaging [34].

Additionally, participants appreciated that the infographic highlighted both successes and areas for improvement; a strategy endorsed by the WHO [28] as part of effective surveillance communication. Incorporating surveillance data into training and education programs within the healthcare institution was also seen as a way to foster accountability, motivation, and continuous improvement.

This perspective aligns with Ajzen’s Theory of Planned Behavior [35], which posits that behavioral intention is influenced by attitudes toward the behavior, perceived social norms, and perceived behavioral control. According to Ajzen [35], attitudes emerge from the beliefs people hold about the behavior, formed by associating it with certain attributes or outcomes. In our study, the use of infographics appears to have fostered positive attitudes among IPC link nurses by making HAIs surveillance data more accessible, engaging, and relevant to their professional context, thereby enhancing their willingness to engage with the data. Perceived social norms refers to the influence of referent individuals or groups who approve or disapprove of a behavior [35]. In our findings, team dynamics and peer engagement emerged as influential factors: participants noted that reviewing and discussing infographic feedback with colleagues reinforced normative expectations of data engagement and increased pressure to act. Perceived behavioral control relates to the perceived ease or difficulty of performing the behavior, reflecting prior experience and anticipated obstacles [35]. Here, the simplicity and clarity of the infographic format reduced interpretive barriers and increased nurses’ confidence in interpreting and acting upon surveillance findings, thereby enhancing perceived control. Together, these elements suggest that infographics may increase intentions to engage with data, thereby supporting the transition from intention to behavior and contributing to improved infection prevention and control practices.

#### Infographics as Catalysts for Quality Improvement Initiatives

Some participants reported initiating quality improvement projects after engaging with the infographic, suggesting that visual tools can act as triggers for behavioral change. Fogg’s Behavior Model [36] posits that behavior occurs when the following three elements converge: motivation, ability, and triggers. A trigger may take many forms, such as the following: notifications, messages, visual prompts, or infographics, as noted by Siricharoen & Siricharoen [9]. In our study, motivation is supported through perceived relevance and anticipated benefits of the surveillance data. Ability is enhanced by the visual format’s clarity and simplicity, which reduce interpretative barriers. Prompts, such as reminders, meetings, or visual cues, act as triggers prompting nurses to consult feedback reports. When motivation, ability, and prompts align, behavioral engagement becomes more likely. Thus, by attending to these elements, infographics may not only increase intention—in line with Ajzen’s Theory of Planned Behavior—but also help actualize behavior, linking intention to action and contributing to improved infection prevention and control practices.

Participants described feeling motivated after interacting with the infographic, noting that key points were better understood and acted upon. To our knowledge, this was the first time healthcare professionals in this setting actively engaged with surveillance data in a way that led to tangible improvements in practice.

When data are made transparent, opportunities for understanding emerge. Infographics foster critical reflection, drive informed action, and effectively connect nursing care with measurable outcomes. Providing nurses with accessible, evidence-based data transcends mere communication, it embodies a commitment to delivering safer, more intentional and evidence-informed care.

### 4.4. Implications for Practice and Research

The use of infographics in the dissemination of epidemiological surveillance data emerges as a promising strategy for communication with nursing professionals. Maintaining this approach may enhance data accessibility and engagement across clinical teams. However, broader involvement of nursing staff in the co-design of infographics could add value, fostering ownership and relevance, and is recommended as a future strategy.

Investing in a multimodal approach to data dissemination—combining visual tools with educational interventions, digital platforms, and feedback mechanisms—may further strengthen the impact of surveillance communication. Given the robustness of this methodology, exploring its integration into HAIs data dissemination represents a valuable direction for future research.

This study focused on perspectives of infection prevention and control link nurses, who play a key role in data dissemination and team engagement. However, understanding the views of other nursing team members could offer additional insights and potentially reveal alternative strategies not yet considered.

Although the infographic design in this study was guided by Murray et al.’s [20] six key principles, future development could benefit from collaboration with professional designers. As Albers [37] cautions, poor design may lead to misinterpretation or information distortion. Ensuring clarity, accuracy, and visual coherence is essential when translating complex data into accessible formats.

### 4.5. Limitations

Despite the rigorous methodological procedures employed in this study, some limitations should be acknowledged when interpreting the findings. First, the data reflect the perspectives of IPC link nurses within a single hospital setting, and do not include the views of other nurses involved in direct patient care. This context may have shaped participants’ responses and may limit the transferability of the findings to other settings or nursing roles. Second, the study was designed and conducted by the institution’s head nurse for infection prevention and control. Although measures were taken to ensure neutrality and confidentiality, the researcher’s close professional relationship with participants may have influenced the interview dynamics. Nevertheless, this familiarity may also have fostered a climate of trust, possibly enhancing participant openness and contributing to the depth and authenticity of the data. Because of these contextual and relational factors, the predominantly positive perceptions of infographics reported in this study may reflect institutional culture and the proactive role of link nurses rather than attitudes across the wider nursing community. Therefore, caution is warranted when applying these findings to other hospitals, nursing populations, or dissemination contexts. Future research involving a broader range of institutions and professional roles would be valuable to validate and expand upon these findings.

## 5. Conclusions

To effectively reduce healthcare-associated infections, robust IPC programs must include timely and accessible dissemination of surveillance data to all healthcare professionals involved in driving change and sustaining continuous improvement. This study explored the infection prevention and control link nurses’ perspectives on the use of infographics as a communication tool for healthcare-associated infection surveillance data and revealed a generally positive perception of their value.

The findings suggest that infographics enhance data accessibility, comprehension, and engagement, effectively addressing limitations associated with traditional reporting formats. By providing insights into data awareness, team engagement, and practice improvement, the study highlights the potential of infographics to support informed decision-making, motivation, and behavioral change among nurses.

Furthermore, the study emphasizes how infographics may influence behavioral intention by shaping attitudes, reinforcing social norms, and increasing perceived control over data use. When integrated into educational and clinical workflows, infographics can foster data literacy, promote reflection and contribute to a culture of continuous quality improvement in infection prevention and control.

Incorporating visual data communication strategies may therefore serve as a valuable complement to existing reporting practices, enhancing the effectiveness of infection prevention and control programs and supporting evidence-based nursing practice.

## Figures and Tables

**Figure 1 healthcare-13-02961-f001:**
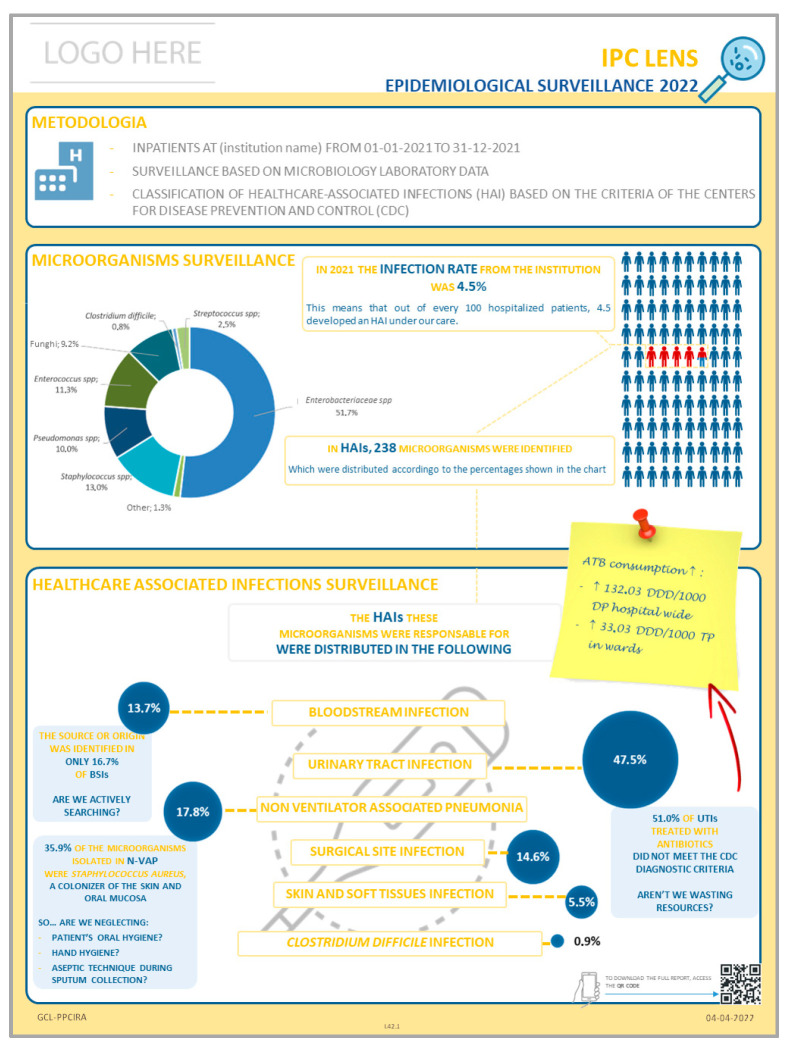
Example of the infographic created to complement the traditional method of HAIs surveillance dissemination.

**Figure 2 healthcare-13-02961-f002:**
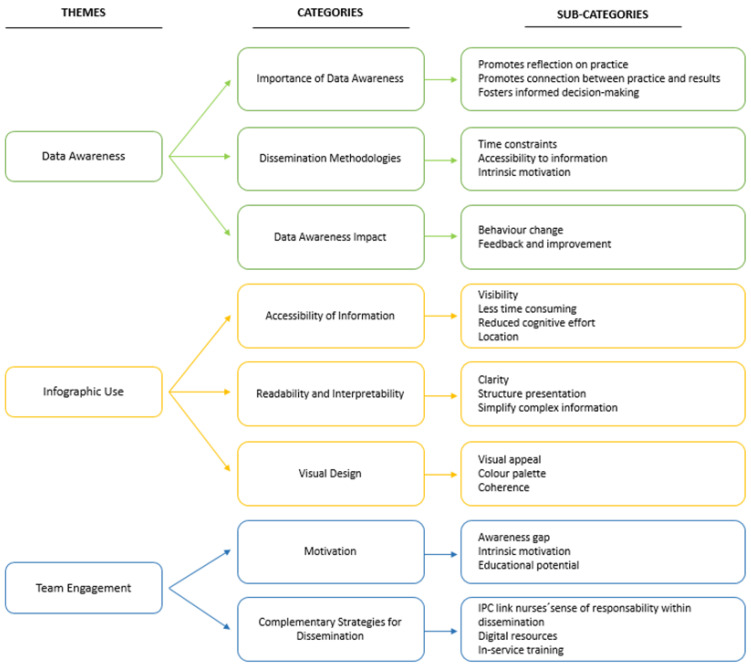
Systematization of nurses’ perception regarding the use of infographics to disseminate HAIs surveillance data.

## Data Availability

For confidentiality purposes, the data are held by the corresponding author (S.F.) and will be made available upon reasonable request.

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
