# Peer review of "The Use of Infographics to Inform Infection Prevention and Control Nursing Practice: A Descriptive Qualitative Study"

_healthcare, 2025, doi:10.3390/healthcare13222961_

Round 1
Reviewer 1 Report
Comments and Suggestions for Authors
Dear Authors.
I am grateful for the opportunity to review your work. After reviewing and analysing your work, I would like to make a number of recommendations and suggestions. I would be grateful if you would take my comments into account as I believe that they can complete your manuscript.
Abstract
The abstract perfectly summarises the content of the manuscript, but should be structured into the following sections: background, methodology, results, conclusions and keywords.
Introduction
The introduction clearly and directly presents the problem to be addressed, places the study in context, and highlights the relevance of using infographics, employing appropriate references. However, the inclusion of a study that has used infographics in the field of nursing could be explored in greater depth in order to reinforce the importance of conveying information using this methodology in the professional nursing context.
It is also recommended that the format of citations throughout the manuscript be reviewed. When referring to a specific study—for example, in the introduction (line 39) ‘As Borges (2017) points out’—the year should not be placed in parentheses, but replaced by the corresponding number in the list of references. Similarly, in line 85 (“according to the recommendations of Arcia et al. (2016)”), the year should be replaced by the corresponding reference number. Please review these issues throughout the text to maintain consistency in the citation style.
Metodology
- I believe that the methodology is adequately described and detailed. However, it might be more appropriate for section 2.1, ‘Background,’ which describes the development of the infographics for the study, to be included after the section on participant selection. This would provide a more coherent explanation of how the material used (the infographics) was developed. Likewise, the research questions presented at the end of this section could be integrated into the study design, as they are directly related to it.
- With regard to the exclusion criteria, it would not be necessary to specify them, as it is understood that participants who do not meet the inclusion criteria will not be able to take part in the study.
- With regard to the total number of participants and their average age, this information should be included in the results section, specifically in the description of the sociodemographic variables. In addition, it would be advisable to include the standard deviation along with the average age.
- At the end of section 2.3, ‘Selection of participants,’ the reference to the Declaration of Helsinki should be moved to the section on ethical considerations, as it corresponds to that content. Finally, is there an approval number for the study issued by the Ethics Committee? It would be advisable to include it in the manuscript to complete the ethical information for the study.
Results
The main themes derived from the interviews, as well as the categories and subcategories, are adequately summarised and represented in Figure 2.
Discuss
In relation to the discussion, the results are correctly analysed and interpreted in light of previous studies. The conclusions and implications are adequately described, and the limitations identified in the study are included.
It is recommended that abbreviations or acronyms be defined the first time they appear in the text in parentheses and that only the abbreviation be used thereafter (e.g., World Health Organisation [WHO] or Urinary Tract Infection [UTI]).
In summary, congratulations on this enlightening study, which can benefit not only nursing, but also the entire population and various fields, by synthesising information and conveying it effectively through infographics.
Author Response
Thank you for taking the time to review our manuscript. We truly appreciate your valuable feedback and have addressed each comment in detail below. Additionally, we implemented all suggested revisions—highlighted in blue —in the resubmitted file.
Comments 1: Abstract - The abstract perfectly summarises the content of the manuscript, but should be structured into the following sections: background, methodology, results, conclusions and keywords.
Response 1:
We appreciate this suggestion and have revised the abstract accordingly to your suggestion. The suggested sections of background, methodology, results and conclusions were introduced into the abstract (page 1, lines 11, 19, 22 and 26). The background briefly sets the scene, the methodology describes our approach, the results summarize key findings, the conclusions state the implications. All changes are highlighted in blue for your convenience.
With regard to the Keywords, they were already included on the same page; however, if you believe modifications are needed (e.g., adjustments to wording, formatting, or order), we would be happy to revise them accordingly.
Comments 2: Introduction - The introduction clearly and directly presents the problem to be addressed, places the study in context, and highlights the relevance of using infographics, employing appropriate references. However, the inclusion of a study that has used infographics in the field of nursing could be explored in greater depth in order to reinforce the importance of conveying information using this methodology in the professional nursing context.
Response 2:
Thank you for this helpful suggestion. We agree that situating our work within prior studies of infographic use in nursing would strengthen the introduction. Although we conducted a thorough literature search, we found few direct examples of infographics being used in the professional nursing practice as our study proposes. We did identify relevant studies in student-education context and in health-science education more broadly, and we found it useful to include two additional studies. Accordingly, we have revised the introduction to address the reviewer’s suggestions (page 2, lines 55-68):
“Today, infographics are widely used across sectors – including social media, advertising, policy and academic publishing – as a compelling strategy for conveying complex messages [10]. In healthcare, they have gained recognition as tools capable of enhancing the delivery of information to both patients and professionals [5]. Specifically, they support patient health literacy [11,12], improve medical communication [13,14] and bolster competencies in evidence-based practice [15]. Within nursing education, infographics have demonstrated utility as instructional resources in domains such as pharmacology, in care philosophy and basic life support training [16–18]. To our knowledge their use in professional nursing practice and information-conveyance to practicing nurses remains underexplored.
Despite the growing adoption use, there is a paucity of research addressing nurses’ perspectives on the deployment of infographics to disseminate surveillance data on HAIs. The present study therefore aims to explore the perspectives of infection prevention and control (IPC) link nurses regarding the use of infographics to communicate HAI surveillance findings.”
All changes are highlighted in blue for your convenience.
Comments 3: It is also recommended that the format of citations throughout the manuscript be reviewed. When referring to a specific study—for example, in the introduction (line 39) ‘As Borges (2017) points out’—the year should not be placed in parentheses, but replaced by the corresponding number in the list of references. Similarly, in line 85 (“according to the recommendations of Arcia et al. (2016)”), the year should be replaced by the corresponding reference number. Please review these issues throughout the text to maintain consistency in the citation style.
Response 3:
Following your suggestion, we have corrected all in‑text citations to the numeric format. For your convenience, the corrections are highlighted in blue at: page 1 (lines 39 and 45), page 3 (lines 107, 138 and 143); page 4 (line 171), page 12 (lines 538, 559, 562, 563 and 571), page 13 (lines 586, 597 and 629), page 14 (lines 635, 655 and 657), page 15 (lines 690 and 692).
Comments 4: Methodology - I believe that the methodology is adequately described and detailed. However, it might be more appropriate for section 2.1, ‘Background,’ which describes the development of the infographics for the study, to be included after the section on participant selection. This would provide a more coherent explanation of how the material used (the infographics) was developed. Likewise, the research questions presented at the end of this section could be integrated into the study design, as they are directly related to it.
Response 4:
Thank you for this observation. To improve coherence in the Methods section, we have relocated the ‘Background’ content to page 3 (immediately after the ‘Participants’ section) and renamed the sections 2.1. Research Design, and 2.2. Setting and Participants. As you suggested, the research questions have been integrated into section 2.1. Research Design (page 2, lines 84-87). All relevant corrections are highlighted in blue in the revised manuscript.
Comments 5: With regard to the exclusion criteria, it would not be necessary to specify them, as it is understood that participants who do not meet the inclusion criteria will not be able to take part in the study.
Response 5:
Thank you for pointing this out. We agree that it may not be strictly necessary to list separate exclusion criteria when inclusion criteria are clearly defined. In light of your comment, we have removed the exclusion criteria section and kept only the key inclusion criteria, with a brief note that participants not meeting those will not be eligible. Accordingly, we have revised the text to address the reviewer’s suggestions (page 2, lines 97-99):
“Participants were eligible if they held an IPC link‑nurse role, were actively employed in the hospital setting, and expressed willingness to participate. Nurses who did not satisfy these inclusion criteria were considered ineligible.”
All changes are highlighted in blue in the manuscript.
Comments 6: With regard to the total number of participants and their average age, this information should be included in the results section, specifically in the description of the sociodemographic variables. In addition, it would be advisable to include the standard deviation along with the average age.
Response 6:
Thank you for this valuable observation. We agree that reporting only the average age of participants does not fully convey the variability of the sample, hence the standard deviation (SD) provides a quantitative measure of how spread out individual ages are around the mean, offering important context for the reader. Accordingly, we have updated the Results section (sociodemographic variables) to include both the mean age and the standard deviation of participants (page 5, lines 209-212):
“A total of thirteen participants were included in the study. All were female, aged between 30 and 52 years (M = 39.5 years, SD = 6.4 years), and had between 1 and 14 years of experience in the IPC link-nurse role (M = 6.0 years, SD = 3.9 years), representing a broad spectrum of professional expertise.”
All changes are highlighted in blue in the manuscript.
Comments 7: At the end of section 2.3, ‘Selection of participants,’ the reference to the Declaration of Helsinki should be moved to the section on ethical considerations, as it corresponds to that content. Finally, is there an approval number for the study issued by the Ethics Committee? It would be advisable to include it in the manuscript to complete the ethical information for the study.
Response 7:
Thank you for your observation. In accordance to your suggestion, we moved the reference to the Declaration of Helsinki from Section 2.3 (“Selection of Participants”) to the “2.6. Ethical Considerations” section (page 5, lines 206-207).
Furthermore, we confirm that the study was reviewed and approved by the Ethics Committee of the institution under approval number (code 14.OBS.2023, 07 August 2023). This approval number has now been included in the manuscript in the Ethical Considerations section (page 5, line 204).
All changes are highlighted in blue in the revised manuscript.
Comments 8: Results - The main themes derived from the interviews, as well as the categories and subcategories, are adequately summarised and represented in Figure 2
Response 8:
Thank you for your positive feedback regarding the Results section. We are pleased that the main themes derived from the interviews, together with their associated categories and sub‑categories, are clearly summarized and appropriately represented in Figure 2.
Comments 9: Discuss - In relation to the discussion, the results are correctly analysed and interpreted in light of previous studies. The conclusions and implications are adequately described, and the limitations identified in the study are included.
Response 9:
Thank you for this encouraging feedback.
Comments 10: It is recommended that abbreviations or acronyms be defined the first time they appear in the text in parentheses and that only the abbreviation be used thereafter (e.g., World Health Organisation [WHO] or Urinary Tract Infection [UTI]).
Response 9:
Thank you for pointing this out. We have reviewed and updated the manuscript accordingly. Each abbreviation or acronym is now defined in full at first mention with the abbreviation in parentheses (e.g., “infection prevention and control (IPC)” and “World Health Organization (WHO)”), and only the abbreviation is used subsequently. All changes have been highlighted in blue in the revised manuscript:
- The abbreviation WHO has been corrected on pages 3 (line 107), page 12 (lines 527, 549) and page 13 (line 609);
- The abbreviation IPC has been corrected on page 15 (line 714).
5. Additional clarifications
In the course of our review, we came across some other instances where improvements could be made.
We would like to bring to your attention for spelling errors (highlighted in grey): on page 2, line 73 - where we have substituted the word "vonducted" for "conducted".
Finally, we would like to express our sincere gratitude to the reviewer for taking the time to assess our manuscript. Your thoughtful comments provided a valuable opportunity for us to improve the paper, and we greatly appreciate your contribution.
Reviewer 2 Report
Comments and Suggestions for Authors
Thank you for inviting me to review the manuscript titled “The Use of Infographics to Inform Infection Prevention and Control Nursing Practice: A Descriptive Qualitative Study.” This is a potentially interesting and important topic with practical implications for infection prevention and health communication. Below, I provide detailed comments for the authors’ consideration.
Key Issues with the Manuscript
- Aims and Research Questions
- The research questions should be clearly stated in the introduction.
- Please describe how the research questions were developed.
- Methods
- Page 2, line 66: This section would fit better in the Introduction and should be integrated contextually.
- The authors should include a reporting guideline (e.g., COREQ) as an appendix.
- Page 3, line 155: This statement belongs in the Results section.
- Provide more details on how the interview questions were developed and consider including them as an appendix.
- Clarify what probing questions were used during interviews.
- Explain how data saturation was achieved.
- Indicate whether a pilot test of the interview guide was conducted.
- Clarify or describe the relationship between the researcher(s) and participants, including any potential influence on data collection or interpretation.
- Discussion
- Clarity on Limitations and Generalizability:
The authors appropriately note the single-site setting and the potential influence of the researcher’s role as limitations. However, the discussion could be strengthened by explicitly considering how these limitations may have shaped the findings. For example, the predominantly positive perceptions of infographics may reflect the specific institutional culture or the participants’ existing engagement as “link nurses.” Including a statement cautioning against over-generalization to the broader nursing population would enhance transparency. - Theoretical Integration:
The application of the Theory of Planned Behavior (TPB) and Fogg’s Behavior Model is valuable but could be more fully developed.
For TPB, consider explicitly linking the findings to its three constructs:
- Attitudes: How do infographics foster positive attitudes or engagement?
- Subjective norms: How might team engagement or peer dynamics influence behavior?
- Perceived behavioral control: How do infographics enhance confidence in interpreting data or implementing practices?
For Fogg’s Model, the argument that infographics act as “triggers” is plausible, but the discussion of “ability” and “motivation” components needs more depth and clearer linkage to the data.
- Depth on “Team Engagement Awareness”:
Section 4.3 contains valuable insights but overlaps somewhat with 4.1. The unique contribution appears to be the role of infographics in catalyzing quality improvement initiatives. Consider reorganizing or renaming this section to emphasize this finding more clearly.
Typos
Line 466: “a essential” → “an essential.”
Line 513: “accessible” → “accessibility.”
Line 627: “perspective’s” → “perspectives.”
The manuscript would benefit from professional English language editing to improve clarity, flow, and grammar throughout.
Author Response
Thank you for taking the time to review our manuscript. Your insightful comments were invaluable in enhancing its quality. We have responded to each point individually below, and for your convenience, all suggested corrections are highlighted in yellow in the resubmitted file.
Comments 1: Aims and Research Questions - The research questions should be clearly stated in the introduction.
Response 1:
Thank you for your observation regarding the placement of the research questions. We appreciate your emphasis on the importance of clearly stating them in the manuscript. In our revised version, we have placed the research questions within Section 2.1 (Research Design) as part of the methodology, in accordance with the suggestion from the first reviewer. We believe that this placement enhances the logical flow by aligning the research questions directly with our study design. You may find these changes highlighted in blue on page 2 (lines 84-87).
Comments 2: Aims and research questions - Please describe how the research questions were developed.
Response 2:
Thank you for this insightful request. We have added a paragraph in the manuscript explaining how the research questions were developed. Specifically, we conducted a preliminary review of the available literature on infographic use, nurse‑perspectives, and infection prevention data dissemination to identify gaps and inform the focus of our investigation. These gaps led us to frame the research questions around awareness, perceptions and perceived impact of infographics among IPC link nurses. All changes are highlighted in yellow in the revised manuscript, on page 2 (lines 78-84).
“The research questions were developed following a review of relevant literature on infographic usage in healthcare communication and the dissemination of surveillance data among nursing professionals. This review revealed limited evidence regarding the views of practicing nurses on the use of infographics for healthcare‑associated‑infection (HAI) feedback. Based on these identified gaps, and in alignment with the study’s aim to explore the utility of infographics among infection‑prevention and control (IPC) link nurses, we formulated three research questions: (1) Are IPC link nurses aware of HAI surveillance data? (2) What are IPC link nurses’ perceptions of infographic‑based dissemination of HAI surveillance data? (3) What are IPC link nurses’ perceptions of the impact of infographics on feedback of surveillance data and awareness in infection prevention and control?”
Comments 3: Methods - Page 2, line 66: This section would fit better in the Introduction and should be integrated contextually.
Response 3:
Thank you for your thoughtful suggestion regarding the placement of the “Background” section. We appreciate your perspective that it might fit within the Introduction. In our revision, however, following the guidance from the first reviewer, we moved the section to follow “Setting and Participants”, integrating it as part of the methodology. We believe this placement enhances the logical progression from participant context to material development. Nonetheless, to ensure clarity, we have added a brief link in the Introduction noting that further background on the materials used is provided later (page 2, lines 68-70):
“Further details on the development of the infographics are described in Section 2.3 to provide full transparency on the material used in this study.”
All changes are highlighted in yellow for your convenience.
Comments 4: The authors should include a reporting guideline (e.g., COREQ) as an appendix.
Response 4:
Thank you for your valuable suggestion regarding the inclusion of a reporting‑guideline checklist (for example, Consolidated Criteria for Reporting Qualitative Research (COREQ)). While we have already referenced this guideline in the text, we recognise that including the full checklist can improve transparency and completeness of reporting.
In the interest of preserving the manuscript’s conciseness and readability, we have included a statement in the Methods section documenting our adherence to the key COREQ domains. Additionally, we have provided the full checklist as Supplementary Materials, should the Editor prefer.
Therefore, we have revised the manuscript, on page 4, lines 192-193:
“To guide the reporting of this study, we followed Consolidated Criteria for Reporting Qualitative Research (COREQ) [23] (checklist available as Supplementary Materials).”
We hope that this approach strikes a balance between transparency and readability, and aligns with the journal’s standards for methodological reporting.
All changes are highlighted in yellow for your convenience.
Comments 5: Page 3, line 155: This statement belongs in the Results section.
Response 5:
Thank you for your insightful comment. We agree that the statement “In the final phase, coded units and categories were reviewed, refined, and validated in relation to the study objectives.” is more appropriately placed in the Results section. We have therefore moved this sentence to Section 3 (Results), on page 5, lines 213-214, in the revised manuscript:
“Through narrative analysis, coded units and categories were reviewed, refined, and validated in relation to the study objectives, resulting in three main themes.”
All changes are highlighted in yellow for your convenience.
Comments 6: Provide more details on how the interview questions were developed and consider including them as an appendix. Clarify what probing questions were used during interviews. Explain how data saturation was achieved. Indicate whether a pilot test of the interview guide was conducted.
Response 6:
Thank you for your important suggestions. In response, we have reviewed and expanded the section detailing how the interview guide was developed in Section 2.4 Data Collection (page 4, lines 148-160). We have also added a description of the probing questions used during interviews and provided the interview guide as Supplementary Materials. Furthermore, we have included additional detail on how data saturation was determined and clarified that no pilot test of the interview guide was conducted.
“A semi‑structured interview guide was developed to align with the study’s research questions and explore IPC link nurses’ perceptions of infographics for healthcare‑associated infection (HAI) surveillance‑data dissemination. The development process involved a review of relevant literature on infographic usage and data‑feedback practices, followed by the drafting of open‑ended questions and prompts. Although a formal pilot test of the interview guide was not conducted due to time/resource constraints, it was reviewed by a qualitative-research expert and refined accordingly. The final guide covered key domains: awareness of HAI surveillance data; perceptions of infographic‑based dissemination; experiences of current feedback methods; and perceived impacts of the infographics. When participants provided brief responses we followed up with pre-listed prompts – used flexibly - such as: “Could you tell me more about that?”, and “How did you experience that situation?” The complete guide (including main questions and probes) is provided as Supplementary Materials. Data collection occurred between September 2023 and January 2024. Eleven interviews were performed face-to-face in a private meeting room, and two were held via videoconference (Zoom®) due to scheduling constraints. Interviews lasted between 40 and 75 minutes, with a total of 13 interviews conducted. After the 10th interview no new substantive codes or themes were identified in the final three interviews, confirming that data saturation had been achieved. All interviews followed the same guide and were conducted by the first author. Each session was audio-recorded and transcribed verbatim. After transcription, the interview transcripts were shared with the participants (i.e. member checking) to ensure accuracy and allow for clarifications or corrections. Codes and themes generated from the data were subsequently confirmed through triangulation with other researchers, enhancing the trustworthiness of the analysis.The researcher had no direct or hierarchical professional relationship with the participants.”
All changes are highlighted in yellow in the revised manuscript.
Comments 7: Clarify or describe the relationship between the researcher(s) and participants, including any potential influence on data collection or interpretation.
Response 7:
Thank you for highlighting this important aspect of researcher–participant dynamics. We note that we already addressed this issue in the manuscript: the Methods section (on page 5, lines 168-169) states that “the researcher had no direct or hierarchical professional relationship with the participants”. In Limitations section we acknowledge that the principal investigator’s role is the institution’s head nurse for infection prevention and control. By acknowledging the researcher’s positionality, we aimed to enhance transparency and allow readers to assess how the researcher’s background may have shaped data collection, analysis and interpretation. We believe these disclosures provide the necessary transparency about the nature of the relationship and its possible influence on data collection and interpretation. Should the reviewer or Editor prefer further detail (for example, a separate reflexivity paragraph or positionality statement), we are ready to provide it.
Comments 8: Discussion - Clarity on Limitations and Generalizability:
The authors appropriately note the single-site setting and the potential influence of the researcher’s role as limitations. However, the discussion could be strengthened by explicitly considering how these limitations may have shaped the findings. For example, the predominantly positive perceptions of infographics may reflect the specific institutional culture or the participants’ existing engagement as “link nurses.” Including a statement cautioning against over-generalization to the broader nursing population would enhance transparency.
Response 8:
Thank you for bringing this to our attention. We have revised the Limitations section accordingly, clearly linking the identified constraints to their potential impact on the results and cautioning against over‑generalization. All revisions are highlighted in yellow in the revised manuscript for your convenience. The following changes have been made:
- We added the sentence (on page 15, lines 699-701) – “This context may have shaped participants’ responses and may limit the transferability of the findings to other settings or nursing roles”;
- We added the sentence (on page 15, lines 706-710) – “Because of these contextual and relational factors, the predominantly positive perceptions of infographics reported in this study may reflect institutional culture and the proactive role of link nurses rather than attitudes across the wider nursing community. Therefore, caution is warranted when applying these findings to other hospitals, nursing populations or dissemination contexts”.
Comments 9: Theoretical Integration:
The application of the Theory of Planned Behavior (TPB) and Fogg’s Behavior Model is valuable but could be more fully developed. For TPB, consider explicitly linking the findings to its three constructs:
Attitudes: How do infographics foster positive attitudes or engagement?
Subjective norms: How might team engagement or peer dynamics influence behavior?
Perceived behavioral control: How do infographics enhance confidence in interpreting data or implementing practices?
Response 9:
Thank you for the insightful suggestion. We have revised the Discussion section to more explicitly link our empirical findings to the three constructs of the Theory of Planned Behavior—attitudes, subjective norms, and perceived behavioural control—and clarified how each appears in the context of infographic use among IPC link nurses. All changes are now highlighted in yellow the revised manuscript. The following paragraph has been reviewed (page 14, lines 639-651):
According to Ajzen [35], attitudes emerge from the beliefs people hold about the behaviour, formed by associating it with certain attributes or outcomes. In our study, the use of infographics appears to have fostered positive attitudes among IPC link nurses by making HAIs surveillance data more accessible, engaging and relevant to their professional context - thereby enhancing their willingness to engage with the data. Perceived social refer to influence of referent individuals or groups who approve or disapprove of a behaviour [35]. In our findings, team dynamics and peer engagement emerged as influential: participants noted that reviewing and discussing infographic feedback with colleagues reinforced normative expectations of data engagement and increased pressure to act. Perceived behavioural control relates to the perceived ease or difficulty of performing the behaviour, reflecting prior experience and anticipated obstacles [35]. Here, the simplicity and clarity of the infographic format reduced interpretive barriers and increased nurses’ confidence in interpreting and acting upon surveillance findings - thereby enhancing perceived control. Together, these elements suggest that infographics may increase intentions to engage with data, thereby supporting the transition from intention to behaviour and contributing to improved infection prevention and control practices.
Comments 10: For Fogg’s Model, the argument that infographics act as “triggers” is plausible, but the discussion of “ability” and “motivation” components needs more depth and clearer linkage to the data.
Response 10:
Thank you for the insightful suggestion. We have revised the Discussion section to more explicitly link our empirical findings to Fogg’s Behaviour Model and clarified how each element appears in the context of infographic use among IPC link nurses. All changes are now highlighted in yellow the revised manuscript. The following paragraph has been reviewed (page 14, lines 653-665):
Some participants reported initiating quality improvement projects after engaging with the infographic, suggesting that visual tools can act as triggers for behaviour change. Fogg’s Behaviour Model [36] posits that behaviour occurs when three elements converge – motivation, ability and triggers. A trigger may take many forms - notifications, messages, visual prompts, or infographics, as noted by Siricharoen & Siricharoen [9]. In our study, motivation is supported through perceived relevance and anticipated benefits of with surveillance data. Ability is enhanced by the visual format’s clarity and simplicity, which reduce interpretative barriers. Prompts, such as reminders, meetings, visual cues, act as triggers prompting nurses to consult feedback reports. When motivation, ability, and prompts align, behavioural engagement becomes more likely. Thus, by attending to these elements, infographics may not only increase intention – in line with Ajzen’s Theory of Planned Behaviour - but also help actualize behaviour, linking intention to action and contributing to improved infection prevention and control practices.
Comments 11: Depth on “Team Engagement Awareness”:
Section 4.3 contains valuable insights but overlaps somewhat with 4.1. The unique contribution appears to be the role of infographics in catalyzing quality improvement initiatives. Consider reorganizing or renaming this section to emphasize this finding more clearly.
Response 11:
Thank you for your suggestion. We agree that the finding regarding infographics catalysing quality improvement initiatives is distinctive. To maintain coherence with the data structure (which was organized around the major themes that emerged), we have introduced a sub‑section within Section 4.3 “Team Engagement” titled “4.3.1. Infographics as Catalysts for Quality Improvement Initiatives” (on page 14, line 652). This sub‑section draws on the same empirical material but allows us to emphasize this special contribution more explicitly. This change is highlighted in yellow, on page 14.
Comments 12: Typos
Line 466: “a essential” → “an essential.”
Line 513: “accessible” → “accessibility.”
Line 627: “perspective’s” → “perspectives.”
Response 12:
Thank you for bringing this to our attention. For your convenience, we have highlighted this corrections in yellow on pages 12 (line 534), 13 (line 580) and 14 (line 633).
Additionally, we have renamed Section 4.3 to “Team Engagement”, in alignment with one of the key themes derived from the content analysis. We also changed “facilitates” to “facilitated” (on page 13, line 621) to reflect past tense usage.
We would like to express our gratitude once more to the reviewer for taking the time to assess our manuscript. This was an invaluable opportunity for us to improve our research paper.
Round 2
Reviewer 2 Report
Comments and Suggestions for Authors
My final comment is to check the reference format again.
Author Response
Comments 1: My final comment is to check the reference format again.
Response 1:
Dear Reviewer,
Thank you very much for your valuable feedback. I have now carefully reviewed and updated the reference list to ensure that the formatting is consistent and in full line with the journal’s guidelines. I appreciate your attention to this detail and believe the revisions have improved the clarity and professionalism of the manuscript.